# Unveiling the Compositional Ability Gap in Vision-Language Reasoning Model

♠Tianle Li, ♠Jihai Zhang, ♥Yongming Rao, ♠Yu Cheng
♠The Chinese University of Hong Kong, ♥Tencent Hunyuan Research
`tianleli@link.cuhk.edu.hk`
`https://github.com/ltl3A87/ComPA`

## Abstract

While large language models (LLMs) demonstrate strong reasoning capabilities utilizing reinforcement learning (RL) with verifiable reward, whether large vision-language models (VLMs) can directly inherit such capabilities through similar post-training strategies remains underexplored. In this work, we conduct a systematic compositional probing study to evaluate whether current VLMs trained with RL or other post-training strategies can compose capabilities across modalities or tasks under out-of-distribution conditions. We design a suite of diagnostic tasks that train models on unimodal tasks or isolated reasoning skills, and evaluate them on multimodal, compositional variants requiring skill integration. Through comparisons between supervised fine-tuning (SFT) and RL-trained models, we identify three key findings: (1) RL-trained models consistently outperform SFT on compositional generalization, demonstrating better integration of learned skills; (2) although VLMs achieve strong performance on individual tasks, they struggle to generalize compositionally under cross-modal and cross-task scenarios, revealing a significant gap in current training strategies; (3) enforcing models to explicitly describe visual content before reasoning (e.g., caption-before-thinking), along with rewarding progressive vision-to-text grounding, yields notable gains. It highlights two essential ingredients for improving compositionality in VLMs: visual-to-text alignment and accurate visual grounding. Our findings shed light on the current limitations of RL-based reasoning VLM training and provide actionable insights toward building models that reason compositionally across modalities and tasks.

## 1 Introduction

Recent breakthroughs in large language models (LLMs) have shown that strong reasoning capabilities can emerge through RL, as exemplified by GPT-o1 [Jaech et al., 2024] and DeepSeek-R1 [Guo et al., 2025]. These models demonstrate impressive performance on complex multi-step reasoning tasks in the language-only domain, revealing the potential of RL-style post-training to enhance logical and compositional reasoning [Team et al., 2025, Hou et al., 2025, Shen et al., 2025b]. Inspired by these advances, researchers have begun exploring whether similar training paradigms can be extended to VLMs, which integrate visual perception with language reasoning [Zhan et al., 2025, Huang et al., 2025, Hao et al., 2025, Yang et al., 2025, Wang et al., 2025].

Previous attempts to apply RL with verifiable rewards to VLMs have shown promising gains on individual vision-language tasks such as visual math solving and object localization [Shen et al., 2025a, Meng et al., 2025, Pan and Liu, 2025]. However, it remains unclear whether these improvements extend beyond isolated benchmarks to more complex, realistic scenarios that require the integration of multiple reasoning capabilities. While the compositional abilities of LLMs have been increasingly studied in the context of skill composition [Zhao et al., 2024, Xu et al., 2024b], the extent to

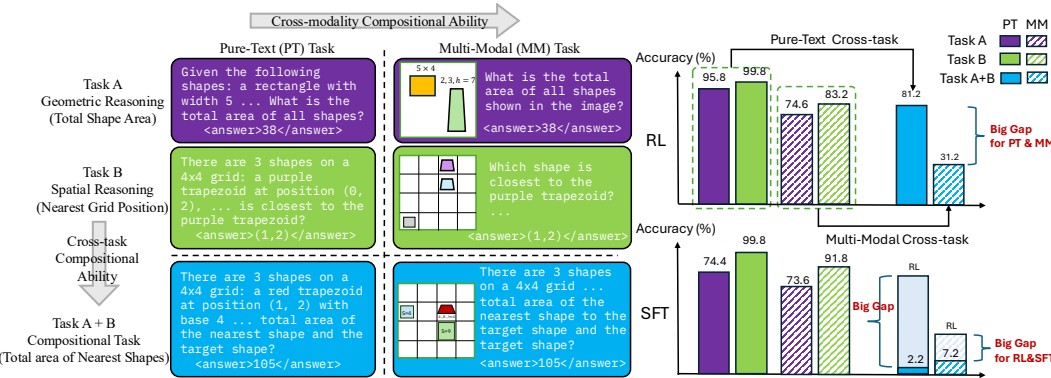

Figure 1: Demonstration of the tasks and partial results for probing of cross-modality and cross-task compositional ability.

which VLMs exhibit similar capabilities remains largely underexplored. In particular, it is still an open question whether VLMs can coherently combine skills acquired independently either across modalities (e.g., transferring textual reasoning to visual inputs) or across reasoning domains (e.g., integrating spatial and arithmetic reasoning) to solve tasks that demand such composition.

To better understand the performance of compositional generalization in VLMs, we investigate two core dimensions: cross-modal and cross-task reasoning. We focus on the following research questions(RQ): **RQ1** Can reasoning abilities acquired through pure-text training be composed with visual recognition to solve multimodal reasoning tasks? **RQ2** Can independently learned visual reasoning skills be integrated to tackle composite tasks that require both capabilities? **RQ3** Can such compositional ability generalize to out-of-distribution (OOD) variants with altered task objectives? To support this investigation, we design a set of diagnostic tasks and carefully curated training and evaluation splits, each aimed at isolating specific challenges such as cross-modal reasoning, visual skill composition, and generalization to new task settings as partially demonstrated in Firgure 1.

We conduct comprehensive experiments across multiple post-training strategies to evaluate the compositional capabilities of VLMs. Our study yields three key observations: (1) RL-trained models consistently outperform SFT in compositional settings, particularly for cross-task generalization; (2) despite strong performance on individual tasks, VLMs exhibit significant limitations in compositional reasoning under multimodal input; and (3) explicitly structuring the reasoning process through visual-to-text prompting and reinforcing intermediate progress reward [Luo et al., 2024] lead to substantial gains in compositional performance.

In a nutshell, our contributions to this work can be summarized as follows:

- We introduce ComPABench, a diagnostic benchmark that systematically evaluates compositional generalization in VLMs across modalities, reasoning tasks, and distribution shifts.

- We conduct a comprehensive empirical analysis of post-training strategies, and reveal their limitations in both cross-modal and cross-task compositional generalization.

- We identify a simple yet effective solution, RL-Ground, that helps reduce the compositional gap in existing post-training strategies by aligning visual inputs to text before reasoning and rewarding accurate grounding of visual content during intermediate reasoning steps.

Together, we hope these contributions can lay a foundation for advancing VLMs toward more robust multimodal reasoning with stronger compositional generalization.

## 2 Related Work

### 2.1 Post-training for VLM Reasoning

Following the success of reasoning-oriented LLMs such as GPT-o1 [Jaech et al., 2024], significant efforts have been devoted to developing advanced reasoning capabilities in VLMs after supervised

fine-tuning over various fundamental visual tasks [Wang et al., 2024a, Chen et al., 2024, Team et al., 2025, Wu et al., 2024, Abouelenin et al., 2025]. Early approaches range from manually designed structured reasoning pathsXu et al. [2024a] to tree-based search strategies Xu et al. [2024a], Yao et al. [2024]. The breakthrough of Deepseek-R1 [Guo et al., 2025] in outcome-based reward RL with GRPO [Shao et al., 2024] for LLMs has inspired attempts to adapt this paradigm to VLMs. However, directly transplanting Deepseek-R1's training methodology to VLMs has proven ineffective. Huang et al. [2025] identify two primary failure modes: (1) the learning algorithm struggles to obtain meaningful positive rewards for complex samples, and (2) models tend to bypass visual inputs and rely solely on textual cues for reasoning. Similarly, Zhan et al. [2025] demonstrate that direct application of outcome reward RL fails to enhance VLMs' reasoning performance. While Du et al. [2025] partially validate the transferability of text-based reasoning capabilities to multimodal tasks, their analysis remains confined to mathematics-oriented domains. Current VLMs, particularly open-source implementations, still exhibit significant gaps in multimodal reasoning compared to human-level performance Hao et al. [2025].

## 2.2 Generalization Probing for Training Strategies

The post-training phase of large models, particularly the choice between supervised fine-tuning and reinforcement learning, has significant implications for generalization. Several recent works have sought to explore these effects for different strategies respectively [Wang et al., 2024b, Zhao et al., 2025]. Chu et al. [2025] conducted a systematic comparison of SFT and RL, finding that SFT often leads to memorization of training patterns, whereas RL enables stronger generalization by encouraging models to discover and apply more transferable principles. Kirk et al. [2023] also confirmed that RLHF-trained models exhibit improved robustness to distribution shifts at the cost of reduced response diversity. In contrast, Yue et al. [2025] revisits the widely held belief that RL with verifiable rewards (RLVR) enables LLMs to acquire fundamentally new reasoning abilities, showing instead that RLVR primarily reweights the model's existing reasoning distribution rather than expanding it, which limits exploratory capacity despite improved efficiency. Our work differentiates from these works by systematically analyzing the limitations of current post-training strategies when applied to VLMs, focusing specifically on their compositional generalization capabilities across modalities, tasks, and out-of-distribution settings.

## 3 Preliminaries

To ground the experimental design and training protocols used in this work, we first formalize the three training paradigms employed throughout our evaluation: Supervised Fine-Tuning (SFT), Reinforcement Learning with Verifiable Reward (RL), and RL training initialized from an SFT-trained checkpoint (SFT-init RL). We also detail the learning objectives used in each setting, with a particular emphasis on the Generalized Reinforcement Policy Optimization (GRPO) strategy adopted in DeepSeek-R1 training [Guo et al., 2025].

### 3.1 Supervised Fine-Tuning

Supervised fine-tuning aligns a pretrained VLM with the target task distribution using paired data $(x, y)$, where $x$ is the input prompt and $y = (y_1, \ldots, y_T)$ is the corresponding output sequence. The model is trained to minimize the negative log-likelihood (NLL) of the target output under the causal language modeling objective:

$$\mathcal{L}_{\text{SFT}}(\theta) = - \sum_{t=1}^{T} \log p_\theta(y_t \mid x, y_{<t}) \tag{1}$$

This objective encourages syntactic correctness and semantic alignment with human-provided examples. In our setup, $x$ may contain text-only format, or both image and text formats. $y$ includes both a reasoning trace (enclosed in a `<think>` block) and a final response (enclosed in a `<answer>` block).

## 3.2 Reinforcement Learning with GRPO

We adopt **Group Relative Policy Optimization** (GRPO) as the optimization strategy in our RL training framework. GRPO generalizes token-level policy optimization with a structured per-sample reward-to-advantage computation and includes a KL regularization term to the reference policy $\pi_{\mathrm{ref}}$ (typically the SFT model).

Let $G$ denote the number of generated candidate answers for a given question $q$, and $|o^{(i)}|$ denote the number of tokens in the $i$-th output $o^{(i)}$. The GRPO loss is defined as:

$$\mathcal{J}\mathrm{GRPO}(\theta) = -\frac{1}{G} \sum_{i=1}^{G} \frac{1}{|o^{(i)}|} \sum_{t=1}^{|o^{(i)}|} \left[ \frac{\pi_\theta(o^{(i)}t \mid q, o^{(i)} < t)}{\pi_\theta(o^{(i)}t \mid q, o^{(i)} < t)} \cdot A(i,t) - \beta \cdot \mathrm{KL}(\pi_\theta \parallel \pi_{\mathrm{ref}}) \right] \quad (2)$$

Here, $A(i,t)$ is the estimated advantage at time step $t$ for the $i$-th sample, and $\beta$ controls the strength of the KL regularization term. The inner term represents a scaled policy gradient with a reward signal modulated at each token step.

In practice, $A(i,t)$ is typically derived from a scalar reward $r(q, o^{(i)})$ measuring task success, which may combine:

- **Answer correctness:** Whether the generated prediction matches the ground-truth answer.
- **Format adherence:** Whether the output satisfies the specified constraints of format.

This formulation ensures that the model improves generation quality while maintaining consistency with the reference distribution.

## 3.3 SFT-Initialized RL Training

To stabilize and accelerate the RL training process, we explore a hybrid strategy where reinforcement learning is initialized from a model pretrained with SFT, similar to R1Guo et al. [2025]. In this setting, the reference policy $\pi_0$ used in the KL term is set to the SFT-trained model, and the initial parameters $\theta_0$ of the policy $\pi_\theta$ are inherited from the same checkpoint. This strategy offers two key benefits: (1) it enables faster convergence by leveraging prior alignment to task distributions, and (2) it mitigates early-stage instability common in pure RL setups.

Together, these training paradigms define the backbone of our experimental pipeline, enabling us to probe the strengths and failure modes of VLMs in compositional, multimodal, and generalization-intensive reasoning settings.

# 4 Experiments

To assess compositional generalization in VLMs under different post-training strategies, we conduct experiments from three perspectives: cross-modal composition, cross-task reasoning, and out-of-distribution generalization. We first present **ComPABench**, our diagnostic benchmark, followed by training setups and evaluation results addressing the three research questions.

## 4.1 Benchmark for Probing of Compositional Ability

| Task Type | Training Subset | Test Subset |
|---|---|---|
| Cross-Modal Composition | PT-GR, PT-SR | PT-GR, PT-SR, MM-GR, MM-SR |
| Cross-Task Composition (Pure-text) | PT-GR, PT-SR | PT-GR, PT-SR, PT-Comp |
| Cross-Task Composition (Multimodal) | MM-GR, MM-SR | MM-GR, MM-SR, MM-Comp |
| OOD Composition | MM-GR, MM-SR | MM-GR-OOD, MM-SR-OOD, MM-Comp-OOD |

Table 1: Statistics of our proposed ComPABench for different task settings. The abbreviations of different types of dataset can be found in the right bottom of blocks in Figure 2.

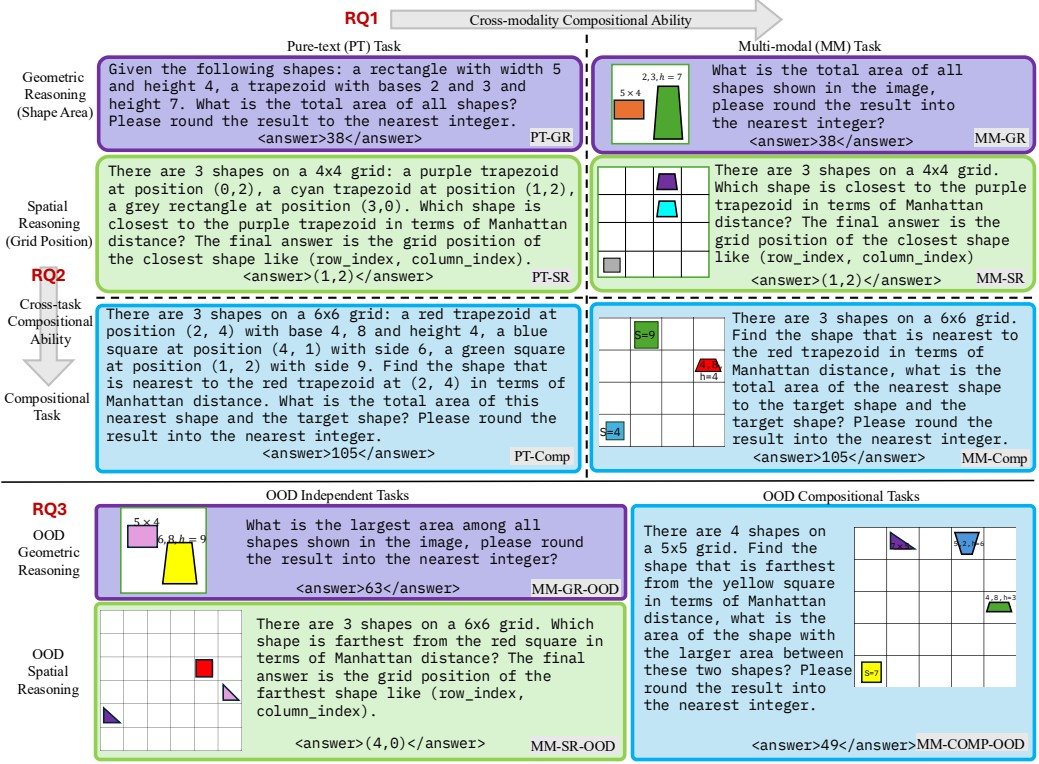

Figure 2: Demonstration of proposed ComPABench for RQ1, RQ2, and RQ3.

To systematically evaluate the compositional ability of VLMs trained under different post-training strategies, we design a set of finely controlled tasks aligned with our three core research questions, where we call this benchmark as **ComPABench**. Each task setting is implemented with paired pure-text and multimodal variants to allow cross-modality and cross-task comparisons. Figure 2 illustrates the benchmark design across individual and compositional tasks, together with the corresponding OOD variants for evaluating transfer robustness.

**Cross-Modal Compositional ability (RQ1).**   To evaluate whether reasoning abilities acquired from pure-text training can transfer to visual inputs at inference, we construct parallel task formats with matched semantics but differing input modalities. In the geometric reasoning task, the model computes the total area of multiple shapes described either in pure-text or shown in an image with labeled dimensions. In the spatial reasoning task, it identifies the grid index of the shape closest to a given target, based on either textual position descriptions or an image depicting a grid with embedded shapes. By comparing performance across modalities, we assess the model's ability to compose textual reasoning with visual perception.

**Cross-Task Compositional ability (RQ2).**   We probe whether models can integrate independently acquired skills by composing geometric and spatial reasoning in a single task. Here, we evaluate whether models trained on each skill individually can solve questions requiring both, such as computing the total area of a target shape and the shape closest to the target. We test VLM under both pure-text and multimodal settings to compare the compositional ability of different input types.

**Compositional OOD Generalization (RQ3).**   To evaluate whether compositional reasoning extends to variants of seen tasks, we develop OOD tasks that modify the objective of individual task, so as the compositional ones. For example, instead of asking for the total area, the model must identify the largest area (for geometric reasoning), or select the farthest shape rather than the nearest (for spatial reasoning). In the compositional OOD setting, models are asked to perform combined tasks using these novel objectives (e.g., compute the area of the larger of between the target shape and the farthest shape of it), probing compositional ability in a more challenging setting.

This benchmark provides a unified and controlled evaluation for diagnosing cross-modal and cross-task generalization, as well as robustness to distributional shifts. We present the detailed composition for each of the tasks in Table 1. More specifically, we generate 4K samples for each individual type of data in training and 500 samples for evaluation. For instance, for Cross-Model Composition task, we mix 4K PT-GR and 4K PT-SR data to train a VLM, and test it on 500 PT-GR, 500 PT-SR, etc. The proposed ComPABench directly supports our investigation into the compositional abilities of current reasoning VLMs under different post-training strategies. More details about the construction of ComPABench is provided in the supplementary materials.

## 4.2  Experiment Settings

To evaluate the effect of different post-training strategies on compositional generalization, we conduct all experiments using backbone models: **Qwen2.5-VL-3B-Instruct** and **Qwen2.5-VL-7B-Instruct**. For training configurations, we apply consistent hyperparameters: a per-device batch size of 1, a learning rate of $1e-6$, and a total of 1 training epoch. For RL experiments, we generate 8 completions per prompt in training. And we set the scale before KL divergence constraints to 0, as we observe a dramatic performance degradation with KL divergence in the objective of optimization. All experiments are conducted on 4 NVIDIA H100 GPUs. Our implementation of GRPO is based on the open-source R1-V[1].

## 4.3  Evaluation Result

In this subsection, we present experimental results answering the three research questions introduced previously.

### 4.3.1  RQ1: Cross-Modality Compositional Generalization

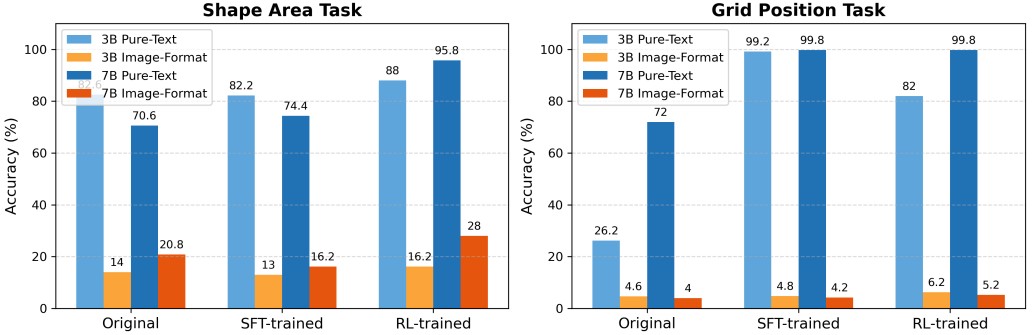

Figure 3: Performance comparison when post-trained with pure-text and evaluated with either pure-text or image-format(multi-modal) questions.

**Pure-text to multimodal generalization gap.** We first evaluate whether reasoning skills acquired from pure-text training transfer effectively to visual input at inference time as shown in Figure 3. While the original Qwen2.5-VL models (without post-training) already show high accuracy on the proposed pure-text tasks, SFT boosts performance in the pure-text modality generally, reaching near-perfect accuracy for the grid position task (99.2% for 3B and 99.8% for 7B), and maintains similar performance or improves moderately for shape area task. However, the models trained solely on pure-text data fail dramatically when tested on the corresponding multimodal tasks, dropping sharply to 13% (3B) and 16.2% (7B) on shape areas, and to just 4.8% (3B) and 4.2% (7B) on grid positions. This large accuracy gap (exceeding 94 points in the worst case) indicates that purely textual training alone does not inherently enable visual reasoning for SFT post-training, despite semantic alignment between tasks.

**Moderate improvement with RL training.** RL generally achieves competitive performance compared to SFT in the pure-text modality after pure-text training, with only one notable exception

---
[1] https://github.com/Deep-Agent/R1-V

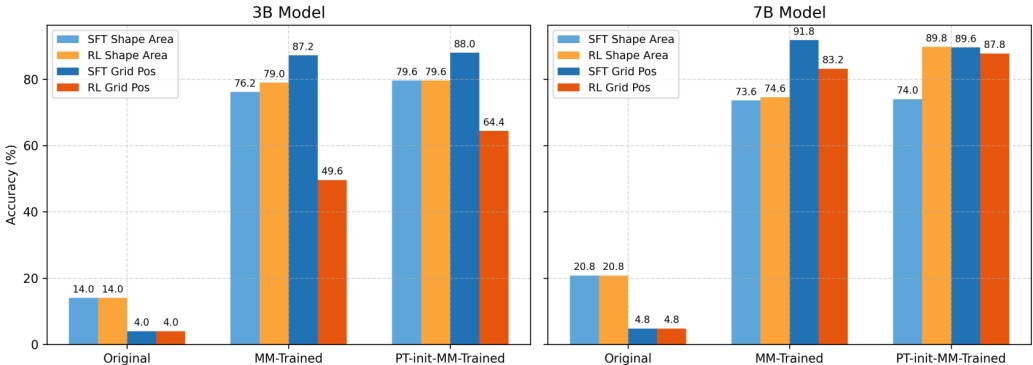

Figure 4: Trend of multi-modal performance without and with pure-text training initialization.

where RL underperforms SFT (82% vs. 99.2% on 3B grid position tasks). Despite this, RL enhances multimodal accuracy compared to pure-text-only SFT models. For instance, RL improves multimodal shape area accuracy from 20.8% to 28.0% (7B), yet still far below the pure-text setting. On the grid-position multimodal task, RL yields only modest improvements (6.2% for 3B and 5.2% for 7B), barely surpassing the performance of the original base model. These results indicate that while RL enables partial compositional generalization from reasoning skills acquired on text to multimodal visual tasks, its effectiveness remains limited when trained exclusively on textual data.

**Impact of initializing multimodal training with pure-text priors.** To better understand how pure-text training influences subsequent multimodal reasoning, we further examine models initialized with pure-text reasoning priors before multimodal training as shown in Figure 4). In this scenario, initializing multimodal RL with models already trained on pure-text data significantly enhances visual task performance. Specifically, for the 3B grid-position task, accuracy increases substantially from 49.6% (direct multimodal RL) to 64.4% (text-initialized multimodal RL). In contrast, the beneficial effect of pure-text initialization is minimal or even slightly detrimental for SFT (91.8% versus 89.6% on Grig Position for 7B model). This discrepancy between RL and SFT likely arises because, during multimodal SFT training, the reasoning path is explicitly provided in the `<think>` block preceding the answer, whereas RL must discover the correct reasoning path solely from final-answer supervision. Therefore, initializing RL with text-trained models, where semantics closely match, may help the model more efficiently converge toward the appropriate reasoning strategy.

These findings collectively confirm that pure-text reasoning capabilities, even when trained to near perfection, do not automatically generalize to multimodal inputs. RL-based training provides moderate improvements over pure-text SFT but is insufficient by itself. Importantly, initializing multimodal RL from a pure-text reasoning model can enhance performance, suggesting an effective strategy for scenarios where multimodal data is limited or costly.

### 4.3.2 RQ2: Compositional Reasoning from Independently Acquired Skills

**Pure-text compositional performance.** In the pure-text setting (left column of Fig. 5), the original Qwen2.5-VL models (without additional post-training) already exhibit moderate compositional capabilities, achieving 49.4% accuracy (3B) and 46.4% accuracy (7B). However, SFT on individual geometric and spatial reasoning tasks separately severely impairs compositional accuracy, dropping performance drastically to just 0.6% (3B) and 2.2% (7B), despite nearly perfect accuracy on each sub-skill individually. This catastrophic forgetting indicates that standard SFT actively disrupts the model's inherent compositional capability. In contrast, RL with a final-answer reward substantially improves compositional reasoning, raising accuracy significantly to 93% (3B) and 81.2% (7B). Thus, RL effectively preserves and enhances compositional generalization in text-only setting for VLMs.

**Multimodal compositional performance.** In the multimodal setting (right column of Fig. 5), the original models (without any multimodal post-training) struggle significantly, achieving low accuracy (5.8% for 3B and 13% for 7B). Similar to the pure-text case, multimodal SFT also fails, reaching only

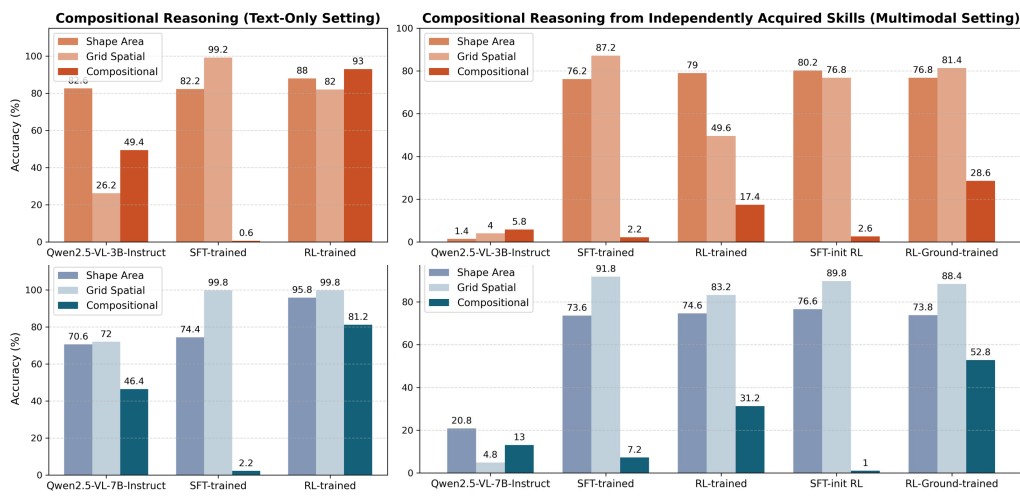

Figure 5: Comparison of Results for compositional reasoning from independently acquired skills.

2.2% (3B) and 7.2% (7B), indicating that standard SFT alone does not enable effective cross-task multimodal compositional reasoning. While multimodal RL training improves upon SFT, achieving 17.4% (3B) and 31.2% (7B), it remains far below pure-text RL levels, highlighting inherent challenges in multimodal compositional reasoning for cross-task scenario.

**Limitations of SFT-init RL.** Initializing multimodal RL training from an SFT checkpoint (SFT-init RL) does not enhance compositional performance; instead, accuracy remains extremely low (2.6% for 3B, dropping to 1.0% for 7B). This indicates that RL struggles to correct flawed compositional strategies established by prior SFT training. We attribute this issue primarily to our hybrid training strategy, which alternates evenly between SFT and RL updates (half-half step attribution). Although SFT-init successfully boosts RL performance on individual tasks as expected due to explicit reasoning path supervision during SFT, it simultaneously imposes strong biases that limit RL's flexibility in adjusting compositional strategies. Consequently, while SFT-init can facilitate faster learning of individual skills, it may inadvertently hinder compositional generalization across tasks compared to RL trained without SFT initialization.

**Impact of progress-reward grounding (RL-Ground).** Motivated by these failures, we explore a strategy that explicitly targets visual-to-text alignment and reasoning decomposition. Our proposed potential solution, **RL-Ground**, combines two key components: (1) a `<caption>` block that forces the model to first describe visual content in natural language before entering the reasoning stage, and (2) a fine-grained progress reward that provides supervision at the level of intermediate vision-grounded reasoning (e.g., correct shape area computation or distance estimation), rather than only at the final answer.

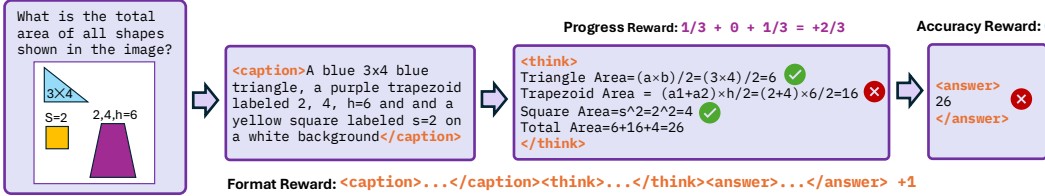

Figure 6: Illustration of RL-Ground framework.

As shown in Fig. 6, the `<caption>` module promotes an early transformation of visual inputs into language, triggering the reasoning together with text instead of merely visual inputs. Simultaneously, the progress reward allows the model to build up compositional reasoning over verifiable subgoals, reducing reliance on sparse final accuracy reward signals. This structure leads to significant gains: RL-

Ground achieves 28.6% (3B) and 52.8% (7B), surpassing the other evaluated post-training strategies. We provide more ablations and training progress on RL-Ground in the supplementary materials.

These findings clearly illustrate that cross-task compositional generalization is challenging under multimodal settings. Simple exposure to independently trained sub-skills via standard SFT or RL alone proves insufficient, and even harmful (in the case of SFT). Instead, caption-before-think formats coupled with dense progress reward significantly improve multimodal compositional reasoning, offering a promising direction for future training paradigms.

### 4.3.3    RQ3: Generalization to OOD Compositional Task

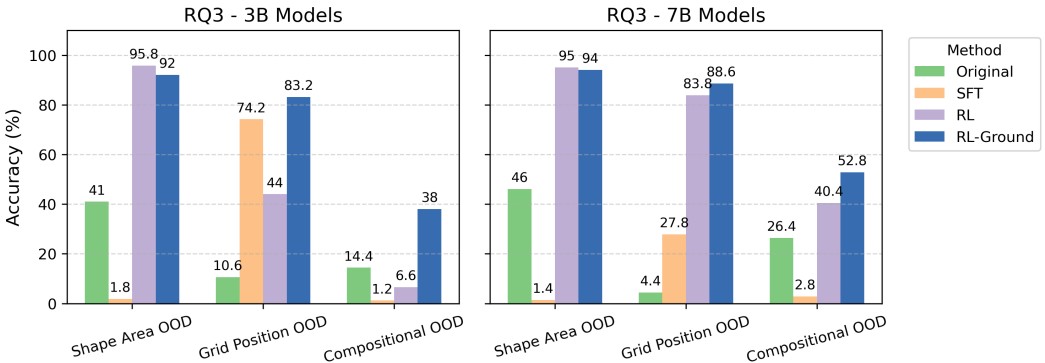

Figure 7: Comparison of Results for generalization to OOD independent and compositional tasks.

**SFT vs. RL under OOD generalization.** We evaluate model robustness on OOD tasks that modify the reasoning objective while maintaining similar multimodal inputs. Results show that SFT exhibits task-dependent generalization: it completely fails on the largest-area task (1.8% for 3B and 1.4% for 7B), but achieves moderate accuracy on the grid position of the farthest shape task (74.2% for 3B and 27.8% for 7B), suggesting limited transferability when visual structures closely align with training data. However, it fails on the OOD compositional task (1.2% for 3B and 2.8% for 7B), reaffirming its lack of compositional flexibility. In contrast, RL generalizes strongly to independent OOD tasks, achieving 95.8% (3B) and 95% (7B) on the largest-area task, and 44% (3B) and 83.8% (7B) on the farthest shape task, comparable to or better than in-domain individual task results from Fig. 5. For the OOD compositional task, RL reveals a scale-dependent trend: while it performs poorly on 3B (6.6%), the 7B model generalizes better, achieving 40.4% and surpassing its in-distribution compositional performance. These results indicate that while SFT's generalization is brittle and task-specific, RL better supports abstract reasoning transfer, particularly at larger model scales.

**RL-Ground achieves robust generalization.** RL-Ground consistently achieves high accuracy across all OOD tasks. For individual OOD task, it outperforms all other methods, reaching 83.2% (3B) and 88.6% (7B) on the farthest shape task(Grid Position OOD). RL-Ground performs merely slightly behind the original RL method in Shape Area OOD task. Notably, RL-Ground also demonstrates best performance on the OOD compositional task, achieving 38% on 3B and 52.8% on 7B model, both matching or exceeding its in-domain compositional performance. These results confirm that combining caption-before-think with progress reward not only enhances in-distribution compositional ability, but also significantly improves robustness to unseen task objectives.

In a nutshell, while SFT exhibits limited and inconsistent generalization under OOD shifts, RL generalizes well to new reasoning objectives, particularly at larger model size. RL-Ground demonstrates the strongest and most stable generalization across all settings, showing clear advantages for both individual and compositional OOD tasks.

### 4.4    Generalization and Grounding Analysis of RL-Ground

While RL-Ground was developed and evaluated on synthetic tasks from ComPABench to facilitate controlled analysis, the techniques it introduces are broadly applicable to real-world VQA settings. In particular, the use of explicit visual grounding via the <caption> step and step-wise progress

reward are not bound to specific data domains. These mechanisms address fundamental challenges in VLMs such as compositional reasoning, grounding robustness, and sparse reward learning.

To validate that RL-Ground is not overly tailored to synthetic setups, we conduct two complementary studies. First, we introduce two auxiliary visual grounding tasks, **Shape Area Grounding** and **Grid Position Grounding**. The two tasks isolate grounding ability from high-level reasoning. In Shape Area Grounding, the model is prompted with a question such as "What is the area of the red triangle?" requiring it to detect the correct shape, recognize its geometry, and compute the corresponding area. Grid Position Grounding challenges the model to locate an object based on color and shape and return its precise row and column index on a grid. Both tasks demand perception and test whether the model extracts the relevant visual facts without performing full reasoning chains.

Table 2 reports results across grounding and reasoning tasks for all evaluated strategies. RL-Ground achieves high scores in both grounding subtasks with 96.2% on Shape Area Grounding and 88.6% on Grid Position Grounding, demonstrating its superior visual grounding alignment via the `<caption>` step and step-wise reward optimization. These improvements directly translate into the highest compositional reasoning accuracy (52.8%), validating the hypothesis that strong grounding enables generalizable multi-hop reasoning.

Interestingly, while standard RL already provides significant gains over SFT with a substantial boost in SA Grounding (96.6% vs. 1.2%) and GP Grounding (74.8% vs. 65.4%), the RL-Ground variant consistently outperforms both, indicating the added benefit of incorporating explicit grounding and structured rewards. This performance trend mirrors the results on the compositional task, where RL-Ground again leads by a wide margin, affirming that improved visual grounding is tightly correlated with stronger generalization and reasoning capabilities.

Table 2: Evaluation of VLMs across grounding and compositional reasoning tasks. "SA" = Shape Area, "GP" = Grid Position. RL-Ground achieves the strongest visual grounding and generalization.

| Model | SA | SA Grounding | GP | GP Grounding | Compositional |
|---|---|---|---|---|---|
| Qwen2.5-VL-7B-Instruct | 20.8 | 89.4 | 4.8 | 24.4 | 13.8 |
| SFT | 73.6 | 1.2 | **91.8** | 65.4 | 7.2 |
| RL | 74.6 | **96.6** | 83.2 | 74.8 | 31.2 |
| SFT-init-RL | **76.6** | 1.2 | 89.8 | 43.6 | 1.0 |
| RL-Ground | 73.8 | 96.2 | 88.4 | **88.6** | **52.8** |

Beyond ComPABench, we assess whether RL-Ground generalizes to real-world language patterns. We perform a zero-shot evaluation on the Geometry3K dataset [Lu et al., 2021], which is a natural language-based VQA benchmark involving spatial and geographic reasoning. RL-Ground, trained only on synthetic data, achieves 21.2% accuracy, outperforming the base Qwen2.5-VL-3B-Instruct (14.8%), SFT (9.2%), and RL-only (17.8%) models. This demonstrates that caption-before-reasoning and progress reward confer robustness that transfers across tasks and domains.

## 5   Conclusion

We introduce ComPABench, a benchmark for evaluating compositional ability in VLMs across cross-modal, cross-task, and OOD settings. Inspired by recent RLVR progress in language models, we assess whether similar training strategies improve compositional reasoning in VLMs. Through comparisons of SFT, RL, and SFT-initialized RL, we find that RL better integrates independently learned skills, especially in cross-task and OOD scenarios. Yet, compositional reasoning with visual inputs remains challenging. Our proposed RL-Ground strategy, combining caption-before-reasoning and progress rewards, yields strong in-distribution and out-of-distribution gains in terms of compositional ability across tasks, underscoring the value of structured prompting and grounded supervision for improving the compositional generalization of multimodal reasoning. To further understand its effect, we evaluate RL-Ground on auxiliary visual grounding tasks and find that it substantially improves grounding fidelity compared to baselines, which in turn correlates with higher compositional performance. Moreover, RL-Ground generalizes effectively to real-world visual question answering, despite being trained only on synthetic data. These findings highlight the transferability of visual-grounded RL and point to new directions for robust multimodal reasoning.

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
