# OpenReview forum: "Unveiling the Compositional Ability Gap in Vision-Language Reasoning Model"
_NeurIPS.cc/2025/Conference — NeurIPS 2025 poster_

### Official Review · Reviewer_JamV · 2025-06-27

**Clarity:** 3
**Significance:** 2
**Originality:** 3
**Rating:** 4
**Confidence:** 4

**Summary:**

This paper investigates the compositional reasoning capabilities of Vision-Language Models, exploring why skills learned independently often fail to integrate in complex, multimodal tasks. To systematically probe this "compositional gap," the authors introduce ComPABench, a diagnostic benchmark designed to test cross-modal and cross-task generalization. The paper proposes a novel and effective solution, RL-Ground, which enhances compositionality by enforcing a "caption-before-thinking" step and rewarding intermediate reasoning progress, leading to significant gains on both in-distribution and out-of-distribution tasks.

**Questions:**

1. The RL-Ground method seems tailored to your synthetic benchmark. How can its "progress reward" be scaled to complex real-world tasks where sub-goals are not easily verifiable? Furthermore, can you provide an error analysis on the brittleness of the "caption-before-thinking" step, particularly when the initial caption is flawed?
2. The paper's conclusions are based solely on the Qwen architecture. What is your argument for why the identified "compositional gap" and the relative performance of SFT vs. RL are fundamental issues that would generalize to other VLM families (e.g., Gemma, Phi-4)?
3. Given that ComPABench is synthetically generated, its linguistic diversity may be limited. Have you tested your RL-Ground model on any existing, more naturalistic VQA datasets to confirm that its reasoning improvements generalize beyond the benchmark's specific language patterns?

**Ethical Concerns:**

["NO or VERY MINOR ethics concerns only"]

**Final Justification:**

Regarding Q1: The new diagnostic experiments are convincing. The data demonstrating that RL-Ground enhances visual grounding effectively mitigates the concern about the brittleness of the <caption> step.

Regarding Q2: I appreciate the authors' candid acknowledgement of this limitation and their commitment to address it in the revised paper. This is a reasonable approach.

Regarding Q3: The new zero-shot evaluation on the GeoQA dataset provides evidence that the method's improvements generalize to realistic, natural-language tasks and are not confined to the synthetic benchmark.

**Limitations:**

Yes.

**Paper Formatting Concerns:**

No.

**Quality:**

3

**Strengths And Weaknesses:**

**Strengths:**
1. The paper introduces ComPABench, a significant contribution. The benchmark is designed to isolate different facets of compositional reasoning, such as cross-modal, cross-task, and out-of-distribution (OOD) generalization, which are often conflated in other evaluations.
2. The study conducts a comparison of post-training strategies (SFT, RL, SFT-init RL) across multiple model scales. The finding that SFT can be actively harmful to compositional reasoning, leading to "catastrophic forgetting" despite high performance on individual skills, is an important result for the community.
3. The paper does not merely identify a problem but also proposes a simple and effective solution, RL-Ground. This method, which combines "caption-before-thinking" with intermediate progress rewards, is intuitive and shows performance gains, offering a promising direction for future research on improving VLM reasoning.

**Weakness:**
1. The experiments are conducted on the Qwen2.5-VL family of models. While using two different model sizes is good for showing scaling effects, validating the findings across different VLM architectures (e.g., Gemma, Phi-4) would significantly strengthen the claim that the "compositional ability gap" is a fundamental issue rather than an artifact of a specific model family.
2. The RL-Ground method relies on providing a "fine-grained progress reward" for intermediate reasoning steps. While this is feasible for the paper's synthetic tasks where ground-truth for sub-steps is clear, defining and verifying these sub-goals for complex, open-ended real-world tasks would be extremely challenging and costly, potentially limiting its practical applicability.
3. The proposed RL-Ground solution relies on the model first generating a caption, but the paper does not explore the failure modes of this approach. If a model generates an incomplete or subtly incorrect caption, it may then reason perfectly from flawed premises, leading to a wrong answer. This "point of no return" at the captioning stage could make the model brittle in real-world scenarios where visual details are ambiguous or hard to describe.
4. Because the ComPABench dataset is synthetically generated, the language of the prompts is likely very structured and repetitive. This lack of linguistic diversity means models may be learning to exploit simple textual patterns rather than developing a robust, generalizable reasoning ability. The conclusions drawn might not fully transfer to real-world datasets where questions are more varied.

---

> ### Author Rebuttal · Authors · 2025-07-30
>
> We thank the reviewer for the constructive feedbacks, and answer the questions raised as follows:
>
> **Q1: Is RL-Ground overly tailored to synthetic tasks? How scalable is progress reward, and how robust is caption-before-thinking to errors?**
>
> We thank the reviewer for raising these important concerns. While our benchmark is synthetic for controllability and diagnosis, the underlying challenges—visual grounding, multi-hop reasoning, and compositional capability—reflect real-world VQA situations. RL-Ground is not tightly coupled to our benchmark; instead, it proposes two broadly applicable directions:
> 1. Enforcing explicit visual grounding (`<caption>`) before reasoning to trigger reasoning with text, and
> 2. Rewarding intermediate progress to encourage visual-grounded reasoning.
>
> **Regarding the generalizability of _progress reward_:**
> We acknowledge that defining sub-goals in open-domain tasks can be challenging. However, many real-world VQA tasks contain implicit intermediate signals (e.g., object detection, caption generation, OCR) that could be repurposed into reward shaping. We envision future work leveraging pretrained vision models (e.g., SAM, GroundingDINO, BLIP-2, YOLO) to estimate such sub-goals in-the-wild as the progress reward.
>
> **On the robustness of the `<caption>` step:**
> We appreciate the concern regarding potential brittleness of the `<caption>` step. To better understand how RL-Ground contributes to compositional generalization, we introduce two supporting grounding tasks that measure the model’s ability to extract critical visual information:
>
> - _Shape Area Grounding_: asks for the area of a specified shape
> - _Grid Position Grounding_: requires locating a shape on the grid using (row, column) indices
>
> These tasks serve as diagnostic tools for analyzing how well the model grounds visual entities before reasoning over them.
>
> Table: Evaluation of VLMs across tasks and their associated visual grounding metrics
>
> | **Model**               | **Shape Area** | **Shape Area Grounding** | **Grid Position** | **Grid Position Grounding** | **Compositional** |
> |-------------------------|----------------|---------------------------|-------------------|------------------------------|--------------------|
> | Qwen2.5-VL-7B-Instruct  | 20.8           | 89.4                      | 4.8               | 24.4                         | 13.8               |
> | SFT                     | 73.6           | 1.2                       | 91.8              | 65.4                         | 7.2                |
> | RL                      | 74.6           | 96.6                      | 83.2              | 74.8                         | 31.2               |
> | SFT-init-RL             | 76.6           | 1.2                       | 89.8              | 43.6                         | 1.0                |
> | RL-Ground               | 73.8           | 96.2                      | 88.4              | 88.6                         | 52.8               |
>
> As shown in the above table, RL-Ground achieves high grounding scores  (_Shape Area Grounding_: **96.2%**, _Grid Position Grounding_: **88.6%**)  compared to all other post-training strategies, along with the best performance on the compositional task (**52.8%**).  This confirms that the `<caption>` step, combined with step-wise reward, enables the model to better structure and retain relevant visual cues, thereby enhancing multi-step compositional reasoning. In contrast to the concern that caption errors could cascade, our results indicate that RL-Ground actually reduces such failure cases by explicitly guiding the grounding process before reasoning begins.
>
> To further mitigate possible risks from incorrect or misleading captions, we identify one promising future direction: introducing a stricter supervision mechanism that treats the caption as a self-contained summary and only assigns reward if the downstream answer can be derived from the caption alone, without using the image. Such an extension would encourage more semantically complete and robust visual summarization. Nonetheless, we emphasize that RL-Ground is not proposed as a final solution, but rather as a light-weight and effective direction that highlights how enhancing atomic visual alignment and grounding can unlock compositional capabilities in VLMs.
>
> **Q2: Does the compositional gap and RL/SFT difference generalize beyond Qwen?**
>
> We believe the observed compositional gap and the relative robustness of RL over SFT are not specific to Qwen, but indicative of broader challenges in current VLM post-training.
> We plan to validate our findings across architectures such as Gemma, InternVL, and Phi-4, as future work.
> To clarify our current scope, we will more explicitly acknowledge the architectural limitation in the paper’s **Limitations** section.
>
> **Q3: Is RL-Ground’s improvement specific to synthetic language patterns? Can it generalize to real-world VQA?**
>
> To assess the generalization beyond our benchmark, we conducted a new zero-shot evaluation on the **GeoQA** dataset (754 test samples), a realistic, natural-language VQA benchmark, using the checkpoint trained with our synthetic dataset and the original  Qwen2.5-VL-3B-Instruct base.
> We evaluated:
>
> - **Qwen2.5-VL-3B-Instruct**: 14.8% accuracy
> - **SFT**: 9.2% accuracy
> - **RL**: 17.8% accuracy
> - **RL-Ground**: 21.2% accuracy
>
> These results confirm that RL-Ground improves performance on naturalistic questions despite being trained only on synthetic data. This highlights that the proposed strategies—captioning and progress reward—enhance reasoning robustness and are not limited to benchmark-specific language patterns.
>
> We will include this experiment and analysis in the revised version to reinforce the practical relevance of RL-Ground beyond ComPABench.

---

> ### Comment · Reviewer_JamV · 2025-08-05
>
> The rebuttal has significantly strengthened the paper's claims and overall contribution. Consequently, I will be raising my score.

---

> > ### Author Response · Authors · 2025-08-05
> >
> > We appreciate the reviewer's constructive reviews and the encouraging follow-up. We are glad we can address the concerns and enhance our work.

---

### Official Review · Reviewer_i1z4 · 2025-07-03

**Clarity:** 3
**Significance:** 3
**Originality:** 2
**Rating:** 3
**Confidence:** 4

**Summary:**

This paper introduces ComPABench, a diagnostic benchmark for systematically probing the compositional generalization ability of Vision-Language Models (VLMs) under different post-training strategies, including Supervised Fine-Tuning (SFT), Reinforcement Learning (RL), and a hybrid SFT-initialized RL approach. The benchmark evaluates cross-modality, cross-task, and out-of-distribution (OOD) compositional generalization. The authors propose a novel RL training strategy, RL-Ground, that incorporates a caption-before-thinking mechanism and progress reward signals to improve performance on compositional tasks.

**Questions:**

Please address the weaknesses above.

**Ethical Concerns:**

["NO or VERY MINOR ethics concerns only"]

**Final Justification:**

Thanks the authors' discussions. I still have some concerns about writing and empirically generality, I'll raise my score to 3.

**Limitations:**

yes

**Quality:**

2

**Strengths And Weaknesses:**

Strengths:
- The paper presents several important observations, e.g., RL tends to preserve compositional skills better than SFT; SFT can overfitting previously acquired reasoning skills.
- The compositional generalization of VLMs is a meaningful research area. This paper targets an important gap in the current research landscape.

Weaknesses:
- The experiments are only conducted on Qwen models of 3B and 7B parameters. This raises concerns about the generality of the findings across model scales and architectures. Larger models (e.g., LLaMA 11B, Gemma 12B/27B) may exhibit different compositional behaviors due to their capacity, and omitting these from evaluation limits the robustness and applicability of the conclusions. While computational cost is understandable, this limitation should be more explicitly acknowledged and, ideally, partially addressed.
- The organization of the paper could be improved. Key components such as the dataset (introduced in the "Benchmark for Probing of Compositional Ability" subsection) and the proposed method (described in the "Impact of progress-reward grounding (RL-Ground)" subsection) are both presented under the Experimental section. This conflation of experimental results with dataset/method description makes it harder to clearly distinguish between contributions and evaluations. A more conventional structure separating methodology and experimentation would enhance clarity and readability.
- While the proposed RL-Ground strategy is practical and achieves notable improvements, its core components—captioning before reasoning and progress-based intermediate rewards—are relatively standard and lack significant innovation. Moreover, RL-Ground fails to address RQ1 (Cross-Modality Compositional Generalization), which is arguably one of the most critical and challenging aspects of VLM compositional reasoning. In addition, the paper does not provide sufficient detail about the experimental setup of RL-Ground.

---

> ### Author Rebuttal · Authors · 2025-07-30
>
> We thank the reviewer for the constructive feedback. Below we address each concern in turn.
>
> **Weakness 1: Limited model scale and architecture diversity**
>
> We acknowledge this limitation and will explicitly state it in the Limitation section. Due to compute constraints, our experiments focus on Qwen models (3B and 7B). Evaluating compositional generalization across other model families such as LLaMA, Gemma, and even closed-source models like GPT-4o is an important direction, and we plan to include these in future work using the same benchmark setup.
>
> **Weakness 2: Organization and motivation of RL-Ground**
>
> We appreciate the suggestion regarding paper organization. Our current structure reflects the probing nature of this work. Our primary goal is to reveal and analyze compositional gaps in MLLMs, a dimension that has received limited attention in prior work. We do not position RL-Ground as a novel algorithmic contribution, but rather as a practical guidepost that highlights how compositional ability may emerge from vision-to-text conversion and visually grounded intermediate reasoning.
>
> We agree that clearer separation of contributions and evaluation will improve readability. In the revised version, we will move benchmark details into a standalone “Benchmark” section and place result analysis, including RL-Ground, into a separate “Results and Findings” section.
>
> **Weakness 3: RL-Ground novelty, RQ1 impact, and experimental details**
>
> We thank the reviewer for this feedback. We emphasize that our goal is not to propose a new algorithm, but rather to highlight two practical directions that improve compositional generalization in RL training:
> (i) explicit conversion from vision to text to trigger subsequent reasoning with text, and
> (ii) incorporating progress-based intermediate rewards to ground visual perception tasks.
>
> Regarding its effectiveness on RQ1 (Cross-Modality Compositional Generalization), we note that RQ1 uses models trained on pure-text corpora, while RL-Ground is a post-training scheme applied to multimodal models. However, to probe the potential transfer of its design principles, we also experimented with applying the `caption-before-think` prompting pattern at inference time during RQ1 evaluation. As shown below, we observe consistent improvements across both tasks:
>
> | Model                       | Shape Area (%) | Grid Spatial (%) |
> |----------------------------|----------------|------------------|
> | RL-3B                      | 16.2           | 4.6              |
> | RL-3B + caption-before-think | 16.6         | 6.2              |
> | RL-7B                      | 28.0           | 3.0              |
> | RL-7B + caption-before-think | 31.4         | 5.2              |
>
> These results suggest that the structured prompting design in RL-Ground may also benefit models evaluated on RQ1, even when trained in purely textual settings.
>
> As for implementation details, RL-Ground uses the same hyperparameter setup as the RL baseline, with two key changes:
> 1. A modified format reward that only gives credit when outputs follow the full `<caption>...</caption><think>...</think><answer>...</answer>` structure.
> 2. An additional progress reward for intermediate reasoning steps, increasing the maximum possible reward from 2 to 3.
>
> We will clarify these implementation specifics in the revised version.

---

> > ### Comment · Reviewer_i1z4 · 2025-08-04
> >
> > Thank you to the authors for the clear rebuttal. The manuscript’s emphasis on revealing generalization deficiencies and providing a diagnostic benchmark is reasonable and worthwhile. That said, I strongly encourage the authors to evaluate GPT-4o with this benchmark; demonstrating limitations in state-of-the-art proprietary models would markedly strengthen the paper’s central claim. I believe this is an interesting and valuable direction of work and encourage the authors to continue refining it. Nonetheless, at this stage I will maintain my original assessment.

---

> ### Author Response · Authors · 2025-08-05
>
> Thank you again for your thoughtful comments and for recognizing the value of our work on diagnosing compositional generalization in MLLMs. We especially appreciate your suggestion to evaluate GPT-4o, and have conducted experiments accordingly.
>
> Below, we compare GPT-4o with Qwen2.5-VL-7B-Instruct (which has been evaluated in the paper) on our benchmark:
>
> | Model                  | Pure-text Shape Area (%) | Pure-text Grid Position (%) | MM Shape Area (%) | MM Grid Position (%) | MM Compositional (%) |
> |------------------------|--------------------------|------------------------------|--------------------|------------------------|------------------------|
> | Qwen2.5-VL-7B-Instruct | 70.6                     | 72.0                         | 20.8               | 4.8                    | 13.8                   |
> | GPT-4o                 | 83.4                 | 97.4                     | 37.0           | 29.6               | 36.0               |
>
> We would like to emphasize that the **focus of our paper is not on evaluating the inherent compositional capability of different models**, but rather on **probing how different post-training strategies influence compositional generalization**. Our experimental design centers on this goal, including the development of a synthetic benchmark that enables controlled training and evaluation of such effects.
>
> In this context, Qwen2.5-VL-7B-Instruct serves as a **zero-shot reference** model without any fine-tuning or post-training on our synthetic datasets. It is used only to illustrate the general trend of performance drop on multimodal compositional tasks when no task-specific post-training is applied.
>
> The results from GPT-4o, while clearly stronger in absolute performance, still show a significant gap between unimodal and multimodal compositional tasks. This supports our central claim: **cross-modal compositional generalization remains an open challenge**, even for top-performing proprietary models.
>
> We would also like to highlight additional insights discussed in our rebuttal to other reviewers, where we conduct fine-grained error analysis on atomic grounding tasks such as area calculation and grid localization for a single shape. These analyses demonstrate that:
>
> - SFT struggles to improve grounding capabilities,
> - RL helps models learn grounding behavior,
> - And our proposed **RL-Ground** further enhances visual grounding and thus improves compositional generalization.
>
> To ensure that our findings generalize beyond synthetic setups, we additionally evaluate our approach on a **realistic benchmark (GeoQA)**, further reinforcing the practical value of our conclusions.
>
> We hope these additional experiments and clarifications demonstrate the strength of our contribution. We kindly invite you to reconsider your assessment in light of these updates.
>
> Thank you once again for your constructive and encouraging feedback.

---

> > ### Comment · Reviewer_i1z4 · 2025-08-08
> >
> > Thanks the authors' discussions. I still have some concerns about writing and empirically generality, I'll raise my score to 3.

---

> > > ### Author Response · Authors · 2025-08-09
> > >
> > > We thank the reviewer again for the constructive and encouraging feedback.
> > >
> > > **On the writing concern.**
> > > We would like to clarify that our writing style follows the conventions of many recent works in *probing analysis*, where the focus is on revealing and analyzing model behaviors rather than introducing a novel method or architecture. This style, as also adopted in [1, 2], emphasizes experimental design and analytical findings over algorithmic innovation.
> > > We are glad that, in your first comment, you noted that *"The manuscript’s emphasis on revealing generalization deficiencies and providing a diagnostic benchmark is reasonable and worthwhile."* Following your suggestion, in the revision we will restructure the paper so that the benchmark/tasks are described in a standalone section, with result analyses placed separately. This is a minor change that can be implemented without significant effort and should directly address your concern about organization and clarity.
> > >
> > > **On the empirical generality concern.**
> > > In response to your suggestion, we have evaluated **GPT-4o** on ComPABench in a zero-shot setting and observed a similar cross-modality compositionality gap as reported in our paper, thereby reinforcing our core conclusion.
> > > Furthermore, to assess **method generalization** beyond synthetic tasks, we have expanded our evaluation to the **realistic GeoQA benchmark** (754 test samples), using the checkpoint trained solely on our synthetic dataset and the original Qwen2.5-VL-3B-Instruct base. Results are as follows:
> > >
> > > | Model                        | Accuracy (%) |
> > > |------------------------------|--------------|
> > > | Qwen2.5-VL-3B-Instruct       | 14.8         |
> > > | SFT                          | 9.2          |
> > > | RL                           | 17.8         |
> > > | **RL-Ground (ours)**         | **21.2**     |
> > >
> > > These results confirm that RL-Ground improves performance on natural-language, real-world VQA questions despite being trained only on synthetic data. This highlights that the proposed strategies enhance reasoning robustness and are not tied to benchmark-specific language patterns.
> > >
> > > We believe these additional results directly address your concern about empirical generality, and we will incorporate them, together with the writing improvements, into the revised manuscript.
> > >
> > > We once again thank the reviewer for the valuable suggestions, which have helped us make the paper more solid and insightful.
> > >
> > > [1] Tianzhe Chu, Yuexiang Zhai, Jihan Yang, Shengbang Tong, Saining Xie, Dale Schuurmans, Quoc V.Le, Sergey Levine, and Yi Ma. Sft memorizes, rl generalizes: A comparative study of foundation model post-training. arXiv preprint arXiv:2501.17161, 2025.
> > >
> > > [2] Yang Yue, Zhiqi Chen, Rui Lu, Andrew Zhao, Zhaokai Wang, Shiji Song, and Gao Huang. Does reinforcement learning really incentivize reasoning capacity in llms beyond the base model? arXiv preprint arXiv:2504.13837, 2025.

---

### Official Review · Reviewer_3k4H · 2025-07-03

**Clarity:** 2
**Significance:** 2
**Originality:** 3
**Rating:** 3
**Confidence:** 3

**Summary:**

This paper investigates the compositionality gap which is a significant limitation of current VLMs about combining cross-modal and cross-task reasoning.
It proposes a newly constructed diagnostic benchmark, ComPABench and finds that while reinforcement learning outperforms supervised fine-tuning in enhancing compositional generalization, models still struggle with complex tasks that require the integration of multiple skills.
To address this, the paper proposes an effective strategy RL-Ground, which significantly improves both in-distribution and out-of-distribution compositional reasoning through a "describe-then-think" approach and a progressive reward mechanism.

**Questions:**

Please add bad cases study and check the training issues.

**Ethical Concerns:**

["NO or VERY MINOR ethics concerns only"]

**Final Justification:**

Following the rebuttal and discussion with the authors, the reviewer remained unconvinced and decided to keep the original rating.

**Limitations:**

Yes

**Quality:**

2

**Strengths And Weaknesses:**

Strength:
- This paper designs and introduces a new ComPABench which is specifically aimed at diagnosing the compositional capabilities of VLMs in cross-modal, cross-task, and out-of-distribution (OOD) settings.
- The paper conducts a comprehensive empirical comparison between two mainstream post-training strategies—Supervised Fine-Tuning (SFT) and Reinforcement Learning (RL) on the benchmark.
- It explores the weaknesses of VLMs in compositionality and proposes a simple solution named RL-Ground. RL-Ground combines a "describe-then-think" prompting structure and achieves significant performance improvements across multiple tasks.

Weakness:

- The paper does not include a qualitative error analysis of model outputs, which would provide deeper insight into the specific failure modes.

- The reported 2.2% performance for SFT on the multimodal compositional task is unconvincingly low, which may suggest a training failure or inappropriate hyperparameters.

- The benchmark tasks are somewhat artificial and detached from practical applications, and the study does not test the models on more realistic multi-task reasoning scenarios.

- Figure 4 inappropriately uses line curves to present different conditions; a bar chart would be a more suitable representation for this data.

- The visualization in Figure 1 is confusing. A clearer approach would be to consistently compare the Pure-Text (PT) and Multi-Modal (MM) results for each of the three task scenarios.

---

> ### Author Rebuttal · Authors · 2025-07-30
>
> We thank the reviewer for their constructive feedback and thoughtful suggestions. Below, we provide responses to the key concerns.
>
>
> **Weakness 1–2: Lack of qualitative error analysis and the low SFT performance on the compositional task**
>
> We thank the reviewer for these important observations. To better understand the behaviors of different post-training strategies, we introduce and analyze two auxiliary tasks:
>
> - **Shape Area Grounding** evaluates whether the model can correctly localize a specific shape and extract its relevant geometric attributes (e.g., base, height) to compute its area. For example, given a prompt like `"What is the area of the purple trapezoid?"`, the task assesses whether the model correctly identifies the purple trapezoid in the image and performs the area computation accordingly.
>
> - **Grid Position Grounding** measures the model’s ability to spatially localize a target shape on a 2D grid. For instance, given the question `"What is the position of the orange right_triangle?"`, the model is expected to identify the correct grid coordinates of the shape in the image.
>
> These grounding evaluations help us identify where failures occur and how they correlate with compositional generalization.
>
> **Table: Evaluation of VLMs across tasks and their associated visual grounding metrics**
>
> | **Model**               | **Shape Area** | **Shape Area Grounding** | **Grid Position** | **Grid Position Grounding** | **Compositional** |
> |-------------------------|----------------|---------------------------|-------------------|------------------------------|--------------------|
> | Qwen2.5-VL-7B-Instruct  | 20.8           | 89.4                      | 4.8               | 24.4                         | 13.8               |
> | SFT                     | 73.6           | 1.2                       | 91.8              | 65.4                         | 7.2                |
> | RL                      | 74.6           | 96.6                      | 83.2              | 74.8                         | 31.2               |
> | SFT-init-RL             | 76.6           | 1.2                       | 89.8              | 43.6                         | 1.0                |
> | RL-Ground               | 73.8           | 96.2                      | 88.4              | 88.6                         | 52.8               |
>
> **SFT overfits to training patterns and fails to adapt to new task prompts**
>
> As shown in the table above, SFT achieves high accuracy on individual reasoning tasks such as Shape Area (73.6%) and Grid Position (91.8%). However, it shows extremely lower grounding scores (1.2% for Shape Area), and its compositional accuracy drops to just 7.2%. This suggests that SFT learns to associate surface-level prompt features with memorized reasoning templates seen during training, rather than learning to flexibly combine reasoning skills.
>
> **Example failure case**
>
> > Question: There are 3 shapes on a 6x6 grid. Find the shape that is nearest to the red trapezoid in terms of Manhattan distance, what is the total area of the nearest shape to the target shape and the target shape?
> > SFT Output:
> > \<think\>Target shape: red trapezoid at (2, 4)
> > Distance from target shape to blue square at (4, 1): 5
> > Distance from target shape to green square at (1, 2): 3
> > Closest shape is the green square at (1, 2) with distance 3
> > Total area of closest shape and target shape: 3 + 8 = 11\</think\>
> > \<answer\>11\</answer\>
> > Ground Truth: 105
>
> In this case, SFT correctly identifies the nearest shape (e.g., the green square) but hallucinates the areas of both the target and the nearest shape, outputting an incorrect total without grounding either value in the actual image. This illustrates that the model overfits to previously seen single-task prompt templates, failing to adapt when the task requires integrating multiple skills or resolving an unfamiliar instruction composition.
>
> **RL improves grounding and partial composition**
>
> Pure RL shows consistently high grounding accuracy (96.6% for Shape Area, 74.8% for Grid Position), enabling better generalization on the compositional task (31.2%). This indicates that RL encourages more faithful visual attention and reasoning behavior. However, the performance gap shows that RL alone is insufficient for strong compositionality without further inductive guidance.
>
> **SFT-init-RL underperforms due to inherited SFT biases**
>
> Although SFT-init-RL performs well on atomic tasks, its grounding scores remain low (e.g., 1.2% for Shape Area Grounding), and its compositional accuracy collapses to 1.0%. This suggests that starting RL training from an SFT checkpoint may preserve the shallow reasoning habits learned during SFT, making it harder for RL to correct them later.
>
> **RL-Ground achieves robust and generalizable behavior**
>
> By enforcing a `<caption>` step and applying step-wise rewards to guide intermediate progress, RL-Ground attains high grounding accuracy (96.2% and 88.6%) and the best overall compositional performance (52.8%). These results offer deeper insight: RL-Ground enhances compositional ability by strengthening the model’s atomic visual grounding capability.
>
> **Weakness 3: Artificial benchmark design and lack of real-world tasks**
>
> We appreciate the reviewer’s concern. The synthetic nature of **ComPABench** is intentional and serves a clear diagnostic purpose: it allows us to precisely control task structures, systematically vary compositional difficulty, and isolate failure modes such as visual grounding, spatial localization, and multi-hop reasoning. While not a replacement for open-domain evaluation, ComPABench complements existing real-world benchmarks by providing a controlled lens to probe compositional reasoning—an ability often underrepresented in current VLM benchmarks.
>
> To assess the generalization of our findings beyond synthetic data, we conducted a zero-shot evaluation on GeoQA, a more natural visual QA benchmark consisting of 754 test samples. Despite being trained only on synthetic data, RL-Ground significantly improves performance:
>
> | Model                      | GeoQA Accuracy (%) |
> |---------------------------|--------------------|
> | Qwen2.5-VL-3B-Instruct     | 14.8               |
> | SFT                       | 9.2                |
> | RL                        | 17.8               |
> | **RL-Ground (ours)**      | **21.2**           |
>
> This evaluation provides three key insights:
>
> 1. **SFT underperforms even the base model**, dropping from 14.8% to 9.2%. This supports our central claim that SFT tends to overfit to synthetic prompt distributions and weakens out-of-distribution generalization.
>
> 2. **RL recovers and surpasses base model performance**, demonstrating that reinforcement learning improves robustness through optimizing the path of thinking more efficiently.
>
> 3. **RL-Ground (ours) achieves the best generalization**, outperforming both RL and SFT. Its design—explicit `<caption>` followed by progress-guided reasoning—encourages the model to extract and structure visual information more effectively, leading to better real-world QA performance even without exposure to in-domain examples.
>
> We will include this GeoQA result and comparative analysis in the revised version to clarify that: While ComPABench is synthetic by design for diagnostic control, the caption-before-thinking strategy and progress-based reward design introduced in RL-Ground can transfer effectively to more naturalistic, multi-task reasoning scenarios, improving model generalization even without training on in-domain examples.
>
> Moreover, we are actively working to extend ComPABench with **real-world image sources** as future plan.
>
> **Weakness 4-5: Visualization issue in Figure 4 and Figure 1**
>
> Thank you for the helpful suggestions. We agree that a bar chart is more appropriate for comparing discrete conditions and will revise Figure 4 accordingly to improve readability. For Figure 1, we will adopt a clearer layout with improved labeling and a legend to better illustrate the task structure and highlight the compositional gap.
>
> **Summary of Planned Revisions**
>
> We thank the reviewer again for highlighting both the strengths and areas for improvement. We will:
>
> - Include qualitative error cases in the final version
> - Clarify the significance of low SFT performance with empirical evidence
> - Revise Figures 1 and 4 for better readability
> - Emphasize that ComPABench is a diagnostic tool, complementary to open-world benchmarks, and reveal the conclusions drawn in our paper are transferable to real-world datasets.
>
> We hope these improvements and clarifications address your concerns and merit a higher score.

---

> ### Author Response · Authors · 2025-08-06
>
> Dear Reviewer 3k4H,
>
> Thank you again for taking the time to review our submission. We deeply appreciate your thoughtful feedback and would welcome the opportunity to continue the discussion. In our rebuttal, we have carefully addressed all the concerns you raised, including detailed error analyses, failure case studies (e.g., on SFT), and the generalization on realistic dataset. We hope that our responses help clarify the contributions and impact of our work more clearly.
>
> As the discussion phase is drawing to a close, we would be grateful if you could take a moment to review our responses and share any further questions or concerns. We would be more than happy to continue the discussion and provide additional clarification if needed.
>
> Thank you for your consideration.

---

> ### Comment · Reviewer_3k4H · 2025-08-07
>
> Thanks for the author's discussions. The reviewer may still have some concerns and decided to maintain their current score.
>
> [w1-w2]
> While the authors provide an example to illustrate the failure mode of SFT, the analysis remains limited in scope and does not convincingly explain the unusually low compositional performance.
> Additionally, the reported behavior of SFT-init-RL appears inconsistent with common empirical patterns observed in similar training paradigms. This raises the question of whether the underlying base model is too weak. Have the authors explored using a stronger base model to verify whether the observed trends still hold?
>
> [w3]
> The reviewer strongly recommend to include these results in the next version of the paper which can strenghthen the experimental completeness.

---

> > ### Author Response · Authors · 2025-08-07
> >
> > We thank the reviewer for the follow-up comments and for clarifying the remaining concerns.
> >
> > **[w1–w2] Clarifying SFT and SFT-init-RL observations**
> >
> > We would like to emphasize that the failure case provided in the rebuttal was meant purely as an illustrative example.
> > Our key supporting evidence for SFT’s unusually low compositional performance comes from the *atomic grounding experiments* on Shape Area and Grid Position. These are much simpler tasks than the full compositional setting, as they only require identifying a single shape’s area or position. SFT performs well on surface-level reasoning for these atomic tasks but fails almost completely on their grounding counterparts (e.g., 1.2% Shape Area Grounding). This shows that SFT has not learned the true reasoning path, but instead relies on superficial prompt–pattern associations, which do not transfer when composition or grounding is required.
> >
> > Regarding SFT-init-RL, as noted in Section 4.3.2 of our submission, our hybrid training strategy alternates evenly between SFT and RL updates (*half–half step attribution*). This design choice differs from prior work, which typically initializes RL from a much earlier SFT checkpoint. Our results demonstrate an *extreme case* where SFT’s overfitting is so entrenched that subsequent RL cannot fully correct it. We also observe that both SFT and SFT-init-RL achieve strong in-domain performance but fails to generalize out-of-domain, reinforcing our interpretation that inherited SFT biases persist through RL in our setting, which do not contradict with previous work with a different setting [1, 2].
> >
> > **[w3] Including additional results**
> >
> > We appreciate the reviewer’s suggestion and are glad to confirm that these results directly address the reviewer’s concern, demonstrating that the conclusions from ComPABench extend to more naturalistic, out-of-domain tasks.
> >
> > We hope this additional clarification addresses the reviewer’s concerns on the scope of our analysis and the reasoning behind the observed trends. If you have further questions or concerns, we are glad to discuss here.
> >
> >
> >
> > [1] Tianzhe Chu, Yuexiang Zhai, Jihan Yang, Shengbang Tong, Saining Xie, Dale Schuurmans, Quoc V.Le, Sergey Levine, and Yi Ma. Sft memorizes, rl generalizes: A comparative study of foundation model post-training. arXiv preprint arXiv:2501.17161, 2025.
> >
> > [2] DeepSeekAI et al. DeepSeek-R1: Incentivizing reasoning capability in LLMs via reinforcement learning,
> > 2025. URL https://arxiv.org/abs/2501.12948. 1, 3, 7

---

### Official Review · Reviewer_tgW9 · 2025-07-04

**Clarity:** 3
**Significance:** 3
**Originality:** 3
**Rating:** 4
**Confidence:** 4

**Summary:**

The paper introduces ComPABench, a carefully-controlled diagnostic suite for probing compositional generalization in vision-language models (VLMs). They evaluate three learning paradigms—SFT, pure-RL with GRPO, and SFT-initialised RL—along cross-modal (text → image), cross-task (integrating spatial + geometric reasoning) and out-of-distribution (OOD) variants. Results on Qwen-VL 3B/7B show: (i) RL consistently beats SFT on compositional tasks; (ii) simply mastering each sub-skill (or pure-text reasoning) does not imply multimodal composition; and (iii) their RL-Ground scheme—“caption-before-think” plus dense progress rewards—substantially narrows the gap, especially on OOD composition.

**Questions:**

- Dataset leakage – Are any ComPABench images or prompts derived from existing public sets? How is overlap with Qwen-VL pre-training filtered?
- Sample size sensitivity – Each test split has 500 items; did the authors compute confidence intervals or run multiple seeds to ensure stability?
- Reward shaping details – Progress rewards are briefly sketched (Fig. 6). Could the authors release the exact reward code and ablative weights?
- Scale trend – Only 3B/7B backbones were tried. Does RL’s advantage persist on ≥32 B models where SFT tends to overfit less?

**Ethical Concerns:**

["NO or VERY MINOR ethics concerns only"]

**Final Justification:**

Thanks the authors for response, I would say the response address most of my questions.

However, after careful considerations and observing other reviewers ratings, I tend to maintain the current ratings due to the significance of this work.

**Limitations:**

- Synthetic, narrow domain – Shapes-on-grid tasks capture discrete arithmetic & spatial reasoning but miss open-world semantics, natural language variety, and real-image noise.
- Small evaluation suite – 500-sample splits and two backbones limit statistical power; no human baselines or comparison to larger proprietary models.

**Paper Formatting Concerns:**

Line 49, “Firgure” typo

**Quality:**

3

**Strengths And Weaknesses:**

- Clear scope – Compositionality is a core open challenge for MLLMs; the benchmark isolates it cleanly.
- Well-designed benchmark – Tasks are paired PT/MM, include controlled OOD splits, and report per-skill vs composed scores, exposing failure modes overlooked by single-task evals.
- Insightful empirical findings – The paper shows RL’s compositional edge and the dangers of SFT “memorisation,” echoing recent LLM studies but in the multimodal space.
- Actionable training recipe – RL-Ground is simple (no new architecture), yet improves both in-domain and OOD performance, offering practitioners a concrete knob to turn.

---

> ### Author Rebuttal · Authors · 2025-07-30
>
> We thank the reviewer for the thoughtful comments and detailed feedback. And we answer the questions raised by the reviewer as follows:
>
> **Q1: Dataset Leakage – Are ComPABench prompts/images derived from public datasets? How is overlap with Qwen-VL filtered?**
>
> All ComPABench images and prompts are synthetically generated in-house using deterministic rendering pipelines (see Supplementary Sec. A). Our datasets are constructed from parametric shape definitions and grid layouts with predefined color palettes, ensuring no overlap with real-world datasets. Therefore, leakage is strictly avoided.
>
> **Q2: Sample Size Sensitivity – Did you compute confidence intervals or run multiple seeds?**
>
> We acknowledge the concern regarding statistical robustness. As noted in the NeurIPS checklist, we ran inference 3 times for all evaluation settings and report average results. We will state it clearly in the revision, which confirms that the key trends remain stable under variance.
>
> **Q3: Reward Shaping – Can the authors release the reward code and ablative weights?**
>
> Yes, we have released the complete reward shaping code in the supplementary zip, and will publicly host it upon acceptance. As detailed in Supplementary Sec. B (Table 2), we conducted ablations isolating the impact of:
>
> - **Caption format only:** Improves grounding but insufficient alone for compositional reasoning.
> - **Progress reward only:** Boosts compositional accuracy from 31.2% to 39.6%.
> - **Full RL-Ground:** Combines both, achieving 52.8% compositional accuracy.
>
> These results validate the design of RL-Ground as both principled and effective.
>
> **Q4: Scale Trend – Does RL’s advantage persist at larger scales (e.g., 32B)?**
>
> Due to compute constraints, our study focuses on 3B and 7B models. While we have not yet evaluated larger models (e.g., 32B), prior work [1] demonstrates that SFT tends to overfit even at 11B, whereas RL provides better generalization under distribution shifts. We leave large-scale evaluation as an important direction for future work.
>
> [1] Tianzhe Chu, Yuexiang Zhai, Jihan Yang, Shengbang Tong, Saining Xie, Dale Schuurmans, Quoc V. Le, Sergey Levine, and Yi Ma. *SFT Memorizes, RL Generalizes: A Comparative Study of Foundation Model Post-Training.* arXiv preprint arXiv:2501.17161, 2025.
>
> **Limitations – Synthetic domain and narrow reasoning types**
>
> We explicitly discuss this in Supplementary Sec. 1. ComPABench is designed as a controlled diagnostic suite (like CLEVR), not as a naturalistic benchmark. While it lacks open-world visual complexity, it systematically isolates reasoning failure modes that broader datasets often conflate. We view it as complementary to large-scale evaluations.
>
> **Typo – Line 49 “Firgure”**
>
> Thank you for noticing. We will correct this in the final version.
>
> In summary, we appreciate the reviewer’s recognition of our benchmark design, training insights, and actionable findings. We hope our clarifications, ablations, and reproducibility efforts address the reviewer’s concerns and justify a higher score.

---

> > ### Comment · Reviewer_tgW9 · 2025-08-04
> >
> > Thanks the authors for response, I would say the response address most of my questions.
> >
> > However, after careful considerations and observing other reviewers ratings, I tend to maintain the current ratings due to the significance of this work.

---

> ### Author Response · Authors · 2025-08-05
>
> We thank the reviewer for the effort in reviewing our paper. We also want to let you know that, as mentioned in the discussion with other reviewer, we also conducted experiments on GeoQA, which is a realistic dataset with natural language diversity.
> - **Qwen2.5-VL-3B-Instruct**: 14.8% accuracy
> - **SFT**: 9.2% accuracy
> - **RL**: 17.8% accuracy
> - **RL-Ground**: 21.2% accuracy
>
> The results demonstrate that our conclusion can generalize beyond our synthetic datasets even without training on the task-specific data. We will include all these clarifications and improvements in our revised version. Thanks again for your recognition of our work.

---

### Note · Authors · 2025-08-12

Our work makes a timely and foundational contribution by being the first to systematically diagnose RL and SFT post-training strategies for multimodal LLMs through the lens of compositional ability, a perspective that is both novel and highly relevant for understanding and improving generalization. The proposed ComPABench offers a controlled framework for probing cross-modality, cross-task, and out-of-distribution compositional reasoning, complementing real-world benchmarks and enabling precise attribution of performance differences to training strategies. Reviewers recognized the value of this diagnostic approach and the meaningful observations it enables, such as RL’s tendency to preserve compositional skills better than SFT. Beyond proposing a benchmark, we also take the first step toward addressing the identified compositional gap by introducing RL-Ground—a simple yet effective method that combines caption-before-thinking and progress reward mechanisms to enhance visual grounding and compositional reasoning. This means our paper not only reveals the weaknesses of current post-training strategies, but also provides an actionable, empirically validated path forward, shedding light on how to improve generalization.

In the rebuttal, we strengthened the paper with new GPT-4o evaluations, confirming that the identified cross-modal generalization gap persists even in state-of-the-art proprietary models, and extended validation to the realistic GeoQA benchmark, where RL-Ground achieves substantial gains over both SFT and RL despite being trained only on synthetic data, demonstrating its generalizability in a zero-shot manner. We also provided fine-grained error analysis on visual grounding tasks, revealing why SFT underperforms relative to RL—offering deeper insight into the mechanisms behind the observed trends. Concerns about model scale, benchmark realism, and paper organization are addressed through explicit acknowledgment, additional experiments, and a commitment to restructuring one section of the manuscript for clarity.

With these enhancements and the adoption of reviewer suggestions, we believe our paper is now both more solid and more insightful, providing a clear and empirically grounded foundation for future research on compositional generalization in multimodal LLMs.

---

### Decision · Program_Chairs · 2025-09-17

**Decision:**

Accept (poster)

**Comment:**

(a) **Summary:**
This paper explores the compositionality gap in VLMs while reasoning, specifically testing whether VLMs inherit compositional abilities during post-training. To benchmark, they introduce a diagnostic suite called ComPABench. Models are first trained on either unimodal tasks or isolated reasoning skills then tested on generalizing to multimodal, compositional variations. They compare SFT and RL settings and find that (i) RL outperforms SFT, showing better integration of learned skills (ii) VLMs perform well on individual tasks yet struggle in cross-modal and cross-task scenarios and (iii) models show better performance in the “caption-then-think” scenario to enforce reasoning.


(b) **Strengths:**
- The paper introduces a new benchmark, ComPABench, and has a very clear scope of testing multimodal compositional understanding (**Reviewer tgW9, Reviewer 3k4H**). There are controlled OOD splits as well as cross-model and cross-task settings.
- The finding that RL preserves compositionality better than SFT is an important and unique finding (**Reviewer i1z4**). Additionally, this setup shows that SFT leads to more catastrophic forgetting compared to RL while maintaining high performance on individual skills (**Reviewer JamV**).
- This work also proposes a solution to the compositionality gap, RL-Ground (**Reviewer JamV**). The idea is intuitive and also leads to improved performance.

(c) **Weaknesses:**
- The authors only tested on Qwen models (**Reviewer i1z4, Reviewer JamV**), which limits the generalizability of the results. Including additional model families and sizes would strengthen the findings.
- ComPABench is synthetic, which means the language is likely more simplistic or repetitive than natural language (**Reviewer JamV**). Similar to only testing on Qwen models, this may limit the generalizability of the findings. It’s unclear if they’ll hold for compositional understanding of natural language.
- The proposed approach of RL-Ground is not novel as the caption first, then reason is used in other paradigms as well (**Reviewer i1z4**). There are also some unexplored failure modes here, such as what happens for incorrectly or poorly generated captions (**Reviewer JamV**).

(d) **Recommendation:** The scores were 3,3,4,4, essentially right at the accept/reject borderline.  The strengths are that this work presents a clean experimental setup alongside a new evaluation suite called ComPABench. ComPABench uses well designed settings (cross-modal, cross-task generalization) and tests RL and SFT settings. Additionally, they propose a new method called RL-Ground to lead to better compositional reasoning. There are also valid concerns, such as only Qwen models being evaluated and the benchmark suite using synthetic language. Overall, weighing the pros and cons, I suggest this paper should be accepted.

(e) **Discussion/Changes during Rebuttal:**
- **Point:** One reviewer questioned the sample size (500 samples) and whether the results are based on multiple seeds/have confidence intervals. The authors ran 3 times and report the average, observing the trends are stable. **Impact:** This experimental setup makes sense and did not impact my decision.
- **Point:** Another reviewer wondered whether RL-Ground is robust to the captioning step (i.e. does a poorly captioned image break the approach). The authors ran an experiment to grounding tasks, evaluating the area and the location of a shape. **Impact:** RL-Ground achieves high performance compared to baseline methods, validating its robustness.